# A Methodological Model to Evaluate Smart City Sustainability

**Alejandro Valencia-Arias** [1,*] , **María Lucelly Urrego-Marín** [2] **and Lemy Bran-Piedrahita** [3]

1   Departamento de Ciencias Administrativas, Instituto Tecnológico Metropolitano, Medellín 050034, Colombia
2   Facultad de Ciencias Económicas y Administrativas, Corporación Universitaria Minuto de Dios,
    Bello 051051, Colombia; murregomari@uniminuto.edu.co
3   Facultad de Ciencias Económicas, Administrativas y Contables, Corporación Universitaria Americana,
    Medellín 050012, Colombia; lbpiedrahita@americana.edu.co
*   Correspondence: jhoanyvalencia@itm.edu.co

**Abstract:** This study aims to identify the key elements that should be included in a methodological model to evaluate the sustainability of smart cities and examines the case of Medellín, Colombia, from 2020 to 2021. For this purpose, it adopted a qualitative descriptive methodology divided into three stages: (1) reviewing existing methodologies for smart city evaluation; (2) designing, validating, and administering information collection instruments; and (3) systematizing and analyzing in-depth interviews. The results indicate that the said model should focus on six key variables: government, mobility, sustainability, people, economy, and quality of life. Smart cities should generate synergies, ensuring the interoperability of their services so that their inhabitants have a better quality of life. The added value of the model proposed here is that it incorporates social and political dynamics, which is noteworthy because most tools in this field do not prioritize them and focus on technical, environmental, transportation, planning, and technological factors.

**Keywords:** sustainability; measurement; smart cities; quality of life; government; mobility; economy

## 1. Introduction

The idea of smart cities has attracted growing interest in the global agenda in the last two decades. The first definitions of smart city emerged after the 1990s as a reflection of the policies different countries adopted to respond to the consolidation of the Information and Communications Technologies (ICTs). Smart cities are technologically modernized territories that, based on techniques supported by smart computing, solve social, economic, and technological issues, which can be seen in the services and infrastructures they develop for their citizens [1].

The progress of ICTs and the changes their introduction has produced in societies have not been in vain because, as a result, new perspectives for the development of the territories have been reconsidered, not only in economic but also social terms. Therefore, in smart cities, the ways citizens interact with the government, the territorial security policies, and even the offer of health care services and the mechanisms to control and monitor diseases have been redefined [2].

In fact, the concept of the smart city has become a key tool that multiple authors have used to address the urban sprawl some countries have experienced in recent years, as well as the consequences derived from this sprawl (e.g., environmental damages). Some models indicate that smart cities should make a contribution at several levels: economic (with innovation and productivity in communities), social (community wellbeing), governance (planning and accountability), and environmental (accessibility and sustainability) [3].

The growth of territories brings more needs and social demands. For the ruling class, this poses a continuous dilemma between satisfying their stakeholders' (citizens') needs and protecting the environment. Thus, it forces them to promote innovative actions that ensure a smart conservation of their resources, that is, adopting sustainable practices [4].

In that sense, sustainable development has been understood as a multidimensional concept supported by the intersection of social, economic, and environmental dimensions. Sustainable development, although discussed since after the Second World War, only gained relevance in the global agenda after the Brundtland Report, where it is defined as "development that meets the needs of the present without compromising the ability of future generations to meet their own needs" [5].

Undeniably, sustainable development gained further relevance after 2015, when the General Assembly of the United Nations (UN) adopted the 2030 Agenda for Sustainable Development in order to transform the reality of the world, especially that of historically marginalized countries. The Agenda is composed of 17 goals divided into five areas: people, planet, prosperity, peace, and partnership [6].

It was defined based on the premise that ensuring poverty eradication requires not only actions regarding economic development but also environmental protection, the improvement of health conditions and education, and fighting historic gender inequality [7]. There are 17 goals in the Agenda: (1) No Poverty; (2) Zero Hunger; (3) Good Health and Well-being; (4) Quality Education; (5) Gender Equality; (6) Clean Water and Sanitation; (7) Affordable and Clean Energy; (8) Decent Work and Economic Growth; (9) Industry, Innovation, and Infrastructure; (10) Reduced Inequality; (11) Sustainable Cities and Communities; (12) Responsible Consumption and Production; (13) Climate Action; (14) Life Below Water; (15) Life on Land; (16) Peace, Justice, and Strong Institutions; and (17) Partnerships to achieve the Goal [8].

This situation calls for studies that investigate smart cities and evaluate their sustainability. Consequently, this paper aims to identify the elements that should be included in a methodological model that can be used to evaluate the sustainability of smart cities; the study case here was Medellín, a city in Colombia, between 2020 and 2021.

This paper is organized as follows: Section 2 details the theoretical background, Section 3 describes the method adopted here, Section 4 reports the main results, Section 5 presents the discussion derived from the information analysis, and Section 6 draws the conclusions.

## 2. Theoretical Background

Sustainable development and smart city, i.e., the main concepts in this study, are discussed below to define the theoretical background of this paper.

### 2.1. Sustainable Development

The definition of sustainable development has changed over time, and has been shaped by different actors interested in the topic such as the government, the business sector, the academy, and civil society [9]. It has been criticized, but it is considered a global issue in the context of the new needs of modern societies. Although it has been addressed throughout history, this concept was consolidated after the Brundtland Report in the 1980s, in which it was defined by the confluence of environmental, economic, and social variables. The guiding principle of sustainable development is meeting human needs while protecting the current and future availability of resources [10].

In this sense, sustainability has been defined as an entity's ability to maintain itself over time. Other approaches in the scientific literature have defined the concept as a dynamic of equilibrium between the satisfaction of human needs and environmental protection; this is in line with the contributions of the Brundtland Report [11].

Other authors have referred to the growing relevance of sustainability in global agendas, which is reflected in the policies and programs several governments have adopted. They involve social, economic, and populational elements, as well as access to basic services (e.g., drinking water and health care) and energy use [12].

The importance of studies about sustainability is justified, among other reasons, by the accelerated demographic growth and the demand for resources that it implies; urban development and migratory movements toward big cities; and, finally, the growing energy consumption of different social actors (i.e., households and the productive sector). There is



no doubt that the operation of big transportation services, for instance, depends on energy, and this clearly poses huge challenges for environmental sustainability [12].

Although smart technologies can act as enablers in the fulfillment of the Sustainable Development Goals (SDGs), two aspects should be taken into account: (1) the research community and the industry have common interests and self-interests in publishing positive results; and (2) if harmful consequences on the environment (i.e., the SDGs in the Environment group) are found, long-term research should be carried out to assess their long-term impact on equity and fairness [13].

However, although sustainability is increasingly important (e.g., its place in the 2030 Agenda), some authors claim that a lack of axiomatic foundations in this field results in the confluence of multiple theoretical positions. Some of those positions refer to the characterization of complex dynamic systems and the management of finite natural resources available (e.g., the capacity of the biosphere to obtain pollutants). This management involves strategies that integrate environmental, social, and economic aspects and should be based on the results of sustainability assessments (focused on recovery and evolutionary processes) to successfully achieve sustainability. Therefore, the discussions about this topic still need new contributions in order to suggest practical applications. Sustainable development clearly reveals a globalized interest in producing a better quality of life for people, thus achieving the sustainability of the human system [14].

### 2.2. Smart Cities

As a result of the growing incorporation of Information and Communications Technologies (ICTs) into societies, new concepts have emerged to represent the current dynamics of life. Smart cities are new types of organizations within specific territories based on urbanization and digitization processes aimed at promoting productivity, competitiveness, and global positioning, among other aspects [15].

In their territorial dynamics, smart cities integrate six concepts: (1) smart economy, with high levels of innovation; (2) smart mobility, based on the adoption of sustainable and eco-friendly transportation systems; (3) smart environment, measured as the adequate management of natural resources; (4) smart communities, based on community training and the development of key skills for innovative ecosystems; (5) smart life, a dimension that can be measured using social indicators that reflect the quality of life of citizens; and (6) smart governance, from the standpoint of the supply of goods and services by government agencies and transparency in public administration [15].

The diversity of positions found in the scientific literature reveals the way ICTs are used to promote new constructs of cities. Smart cities have been called digital cities, information cities, knowledge-based cities, and ubiquitous cities, to mention a few. Therefore, the disciplines that study this topic are increasingly different but are closely related to the achievement of the UN SDGs [16]. Consequently, we reviewed different academic sources to find some key definitions of this concept, which are reported in Table 1.

**Table 1.** Theoretical definitions of smart city.

| Author | Year of Publication | Definition of Smart City |
|---|---|---|
| Harrison et al. [17] | 2010 | Smart cities connect the physical infrastructure, the IT infrastructure, the social infrastructure, and the business infrastructure to leverage the collective intelligence of the city [17]. |
| Caragliuu et al. [18] | 2011 | A city is smart when it invests in human and social capital and traditional (transportation) and modern (ICTs) communication infrastructures to boost sustainable economic growth and generate a high quality of life, with a rational management of natural resources through participatory governance. |

<div align="center">**Table 1.** *Cont.*</div>

| Author | Year of Publication | Definition of Smart City |
|---|---|---|
| Antrobus, Derek [19] | 2011 | The emphasis of smart cities is on the low-carbon economy. For that purpose, they apply policies focused on eco-friendly modernization, which may reduce the greenhouse gas effect and help to identify potential combined energy sources that involve local development. |
| Chourabi et al. [20] | 2012 | The smart city tag is a diffuse concept that is not always used consistently; it may refer to a high-performing progressive city. |
| Bouskela et al. [21] | 2016 | Smart cities put people at the center of development, incorporating ICTs into urban management, and use these elements as tools to stimulate the formation of an efficient government that includes collaborative and citizen planning processes. |

Undeniably, the concept of smart city is related to sustainable development, so much so that the term sustainable city has also been widely studied. As a result, many cities in the world have set themselves the objective of forming the backbone of a large and intelligent infrastructure. Thus, this concept has become a vision, a manifesto, or a promise that, according to Trindade et al. (2017), constitutes the twenty-first century's sustainable and ideal city form based on technologies [22].

## 3. Materials and Methods

This study adopted a qualitative descriptive approach because, as claimed by Patton (2005), this type of research has enabled different authors to address interesting social phenomena from a naturalistic perspective guided by inductive reasoning. Additionally, it allows us to make contributions that offer interpretive richness [23].

Figure 1 details the five stages of this study: (1) reviewing smart city evaluation methodologies found in the scientific literature; (2) selecting a methodology to build information collection instruments; (3) designing and validating the instruments for the fieldwork; (4) administering of the instruments; and (5) systematizing and analyzing the results.

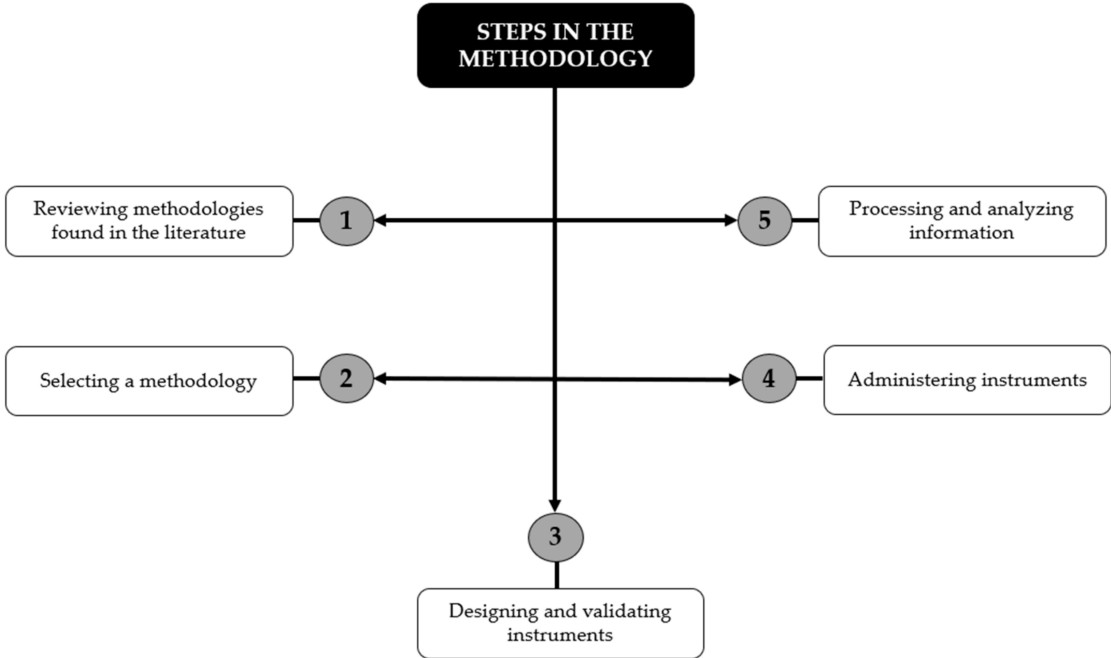

**Figure 1.** Steps in the methodology adopted in this study. Source: Created by the authors.

In the first stage, we reviewed the scientific literature and found a great diversity of methodologies to define the smart city, as well as conceptualizations about the topic, which are summarized in Section 2. For this purpose, we conducted a document search in the Scopus database, which is useful for the systematic collection of scientific high-quality documents. The fields Title and Keywords were used to find the documents to be reviewed. In the second stage, we selected one of those methodologies. The methodology we chose revolves around the dimensions of government, people, economy, quality of life, sustainability, and mobility as key elements to understand the concept of smart cities, which is in line with Öberg et al., 2017 [15]. This methodology was selected here because it is aligned with the orientation of the SDGs, which are holistic goals that should contribute to the development dynamics of countries, especially those that have poor and emergent economies such as Colombia, where this study was conducted.

In the third stage, considering the dimensions that had been previously identified, we designed an information collection instrument, in this case, a semi-structured in-depth interview. This instrument can be used to address elements that have been previously established by the researcher as categories of analysis, but it can be flexible depending on the events that take place during the interviews. Additionally, after the instrument was designed, it was subjected to validation by some key informants in order to ensure its consistency and clarity regarding each one of the questions that were included. The informants were professionals who work in Medellín and who are knowledgeable about the concept of the smart city. Their experience was confirmed based on recent scientific publications and their participation in scientific events.

In the fourth stage, we administered the instrument, interviewing a total of seven academic experts and individuals in the productive sector. The participants were selected by convenience sampling, which is used in qualitative studies when it is difficult to define the profile of the individuals that can meet the requirements of the study [24]. We applied three inclusion criteria: (1) having a basic knowledge of the concept of smart city; (2) living or working in Medellín, Colombia; and (3) having professional and academic experience of five years or longer in any of the six dimensions that define a smart city. Regarding the latter criterion, to ensure the representativity of the sample in qualitative terms, each expert was assigned a dimension according to their academic background, which was confirmed by their scientific publications, the theses they have been an advisor to, and their participation in congresses, seminars, and other events for the social appropriation of knowledge. As a result, the instrument captured the specific narratives of interest in this paper.

This study observed a series of ethical guidelines that ensured the protection of the participants' integrity. They received an informed consent form that detailed the objective, benefits, costs, risks, and other aspects of this study so that their participation was informed and voluntary. Thus, this study complied with standards such as the Declaration of Helsinki, which, although more commonly used in the medical sciences, can be extrapolated to the social sciences, even if it is not an experimental study in which the physical, social, and mental health of individuals is put at risk. In addition, it upheld the principles of non maleficence and oversight of human rights [25].

In the fifth stage, after the interviews were conducted, they were transcribed and coded. For the data analysis, we followed the model by Taylor and Bogdan, which is divided into three phases: (1) data discovery, in which the collected information is initially reviewed and preliminary categories are defined based on the findings; (2) coding, where the data are assigned meaning and grouped into categories identified in the previous stage; and (3) relativization, where researchers consider the context in which the narratives were collected [26]. As a result, a set of categories is revealed in explanatory charts that researchers use to present their findings.

## 4. Results

Based on our analysis, we established the elements that should be included in a methodological model to evaluate the sustainability of smart cities, in particular, Medellín, Colombia; such elements revolve around the challenges of implementing a city model of this kind.

Using the coding that was previously applied and based on the references cited in the Materials and Methods section, the challenges we identified were grouped into six categories: social, economic, political, technological, environmental, and conceptual (Figure 2). The key informants in this study thought that they should be considered in advance in order to design methodologies to evaluate the sustainability of smart cities.

**Figure 2.** Challenges for the implementation of smart city models.

In addition, regarding the key components in a smart city evaluation methodology, the participants mentioned five elements: (1) defining the conceptual framework of the methodology, (2) establishing guiding principles, (3) selecting the analysis approach, (4) establishing indicators, and (5) making a diagnosis of the territory that aims to adopt a smart city model (Figure 3).

The next subsection reports the findings of this study in relation to two elements that have been mentioned before: the challenges and key components of a methodology to evaluate smart city sustainability.

### 4.1. Challenges for the Implementation of Smart City Models

In-depth interviews, conducted with thematic experts in Medellín, revealed a series of challenges for the implementation of smart city models that can be explained by social, economic, political, technological, environmental, and conceptual dimensions (as shown in Figure 2). They should be considered by decision-makers everywhere a model of this kind is implemented.

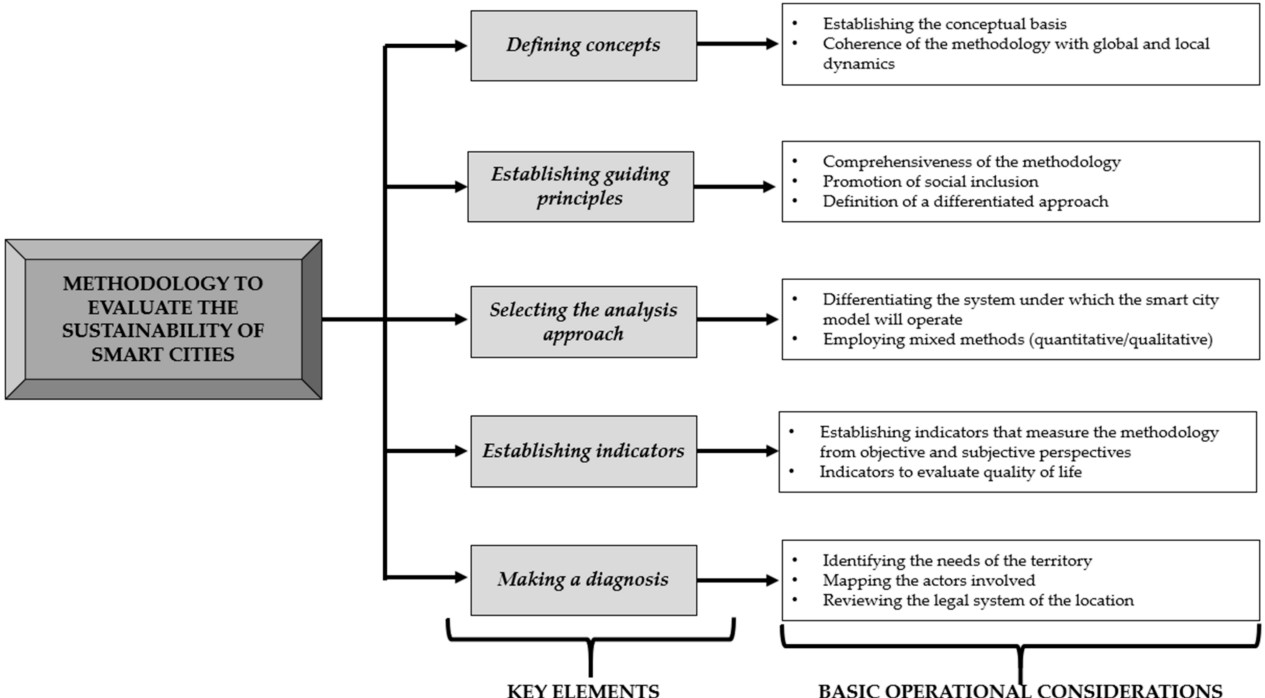

**Figure 3.** Key elements in a methodology to evaluate smart city sustainability.

Regarding social challenges in the context of smart cities, several SDGs are especially relevant: no poverty (SDG 1), quality education (SDG 4), clean water and sanitation (SDG 6), affordable and clean energy (SDG 7), and sustainable cities and communities (SDG 11). This implies, therefore, the creation of circular economies and smart cities that use their resources efficiently [22], for example, low-carbon cities supported by interconnected technologies (e.g., autonomous electric vehicles and smart appliances) [27]. When participants discussed the study case analyzed here (Medellín, Colombia), they referred to the gap between urban and rural regions, unmet basic needs, the promotion of human capabilities, civic commitment, and the influence of cultural patterns.

The participants acknowledged that this gap is a reflection of the general socioeconomic conditions of the country as a whole. This has been widely reported by other authors and international cooperation organizations, who have drawn general attention to the inequality dynamics in Colombia. We consider that the latter are largely associated with the development model of the country and its reflection in the urban sprawl processes of Medellín. In this regard, one of the key informants said,

*"( . . . ) there's still a huge gap that kind of corresponds to the dynamics of this country. It's that, well, the western part of town and El Poblado have undergone a kind of development that is very different from that of the communes up north or Buenos Aires. The form of development is very different in all those places.".* (E-01)

In line with this, the informants in this study also referred to unmet basic needs (UBNs), which they believed should be solved in advance by any city that wishes to migrate to a smart city model. Nevertheless, this criterion should be treated carefully because the context of Medellín is different from that of other capital and mid-sized cities in Colombia. Hence, the participants' claims, instead of being arbitrarily extrapolated to other contexts, reveal a specific reality based on the case studied here. In relation to this, two participants said,

*"( . . . ) from a perspective that's a little utopian, a smart city should be one where, at least, the basic needs of its citizens are met."* (E-01)



*"( . . . ) a city is dynamic, and the entire city does not behave in the same way. I think some places, neighborhoods, commercial places, and industrial areas should be classified by nodes with different needs."* (E-05)

Furthermore, in relation to the UBNs, according to the key informants, social challenges (i.e., access to basic health care services and education) should be considered. This is understandable if we take into account the existing gaps between urban and rural areas in Colombia, as well as this country's historical availability of health care services and educational offers. In this respect,

*"( . . . ) some important things should be solved, for example, security, access to some services, such as health care."* (E-05)

*"( . . . ) smart cities may pose certain challenges related to consumption; labor and occupation; leisure; (very importantly) education; energy transportation with sustainability; and everything related to urban management, health, and well-being."* (E-08)

These social challenges have a core element for the adoption of smart city models: human capabilities. They are increasingly important, even at the national level, as an approach that can guide public policies. The participants mentioned the importance of improving individual capabilities, which requires civic commitment and the influence of cultural patterns; these components were also identified in this study. In this regard, the informants commented,

*"I think developing human potential, an intangible asset, requires adequate spaces for social education and raising environmental awareness."* (E-07)

*"( . . . ) it should be not only a political intention with a deeper knowledge of smart cities but also that of citizens because smart cities need acceptable social behavior to be called that."* (E-04)

*"( . . . ) a pure, radical smart city model can't be applied in some cases. That's the first challenge: trying to deal with and differentiate those cultural characteristics so that any element of a smart city to be applied is feasible for the selected community."* (E-06)

With respect to economic challenges, sustainability has been defined as the equitable satisfaction of the social, economic, environmental, and governance needs of current and future generations. Therefore, for smart cities, widespread economic progress represents a fundamental objective in order to acquire the ability to adapt to change or recover from disruptive events [28]. Our results indicate that the migration to a smart city model should consider the influence of multiple factors: informal labor, migratory processes of vulnerable populations, differentiated local development dynamics, the involvement of business sectors, and the availability of financial resources.

Furthermore, according to the participants, informal labor should be considered part of the reality of Colombia, not only Medellín; it thus constitutes a structural problem. Informal labor could be one of the most crucial economic challenges for the adoption of smart city models because it is related to the national economic policy. In this regard, an informant said,

*"Something I think is a national problem rather than only in Medellín is the high rate of informal labor, which poses another challenge to consolidate a smart city."* (E-01)

Our findings in relation to economic challenges also highlight the impact that migration has had on Colombia in recent years. A high number of migrants have arrived in the country due to complex political and socioeconomic conditions abroad. This is the case in the wave of immigration from Venezuela into Colombia, where Medellín has received one of the highest numbers of foreigners in the country. Additionally, development dynamics have a different influence depending on the territory. The interviewees acknowledged the diversity that can exist in a single geographic space. This aspect can be influenced by other factors that were previously revealed by the participants, such as the gap between urban and rural areas. In relation to these factors,

*"( . . . ) It's different here because it is not so feasible due to economic difficulties, the limited income of many individuals, and the irregular development of our cities, which occurs in waves of people who arrive in urban centers but whose minimum living conditions are not met. As a result, after a certain period, it's back to square one, and that hinders a city from becoming smart."* (E-03)

*"When a city is very dynamic and not all of it behaves in the same way, there are spots, neighborhoods, commercial places, and industrial areas that, I think, should be classified by nodes with differentiated needs.".* (E-05)

We also identified two other factors related to the economic challenges of adopting smart city models. The first one is the productive sector, which the participants in this study considered a key actor because of the role it plays in the transformation of the concept of city, redefining industrial and productive styles. The second factor, crucial from our point of view, is the availability of resources to migrate to this type of model. In the context of Latin American countries (in this case, Colombia), the adoption of smart city models requires not only human talent but also available financial resources to make the necessary changes.

Regarding political challenges, those responsible for designing and promoting creative industry policies for smart cities can achieve positive and more successful results if they pay attention to cultural industries that link industry with society in the relationship of production and consumption [29]. Thus, the participants commented that it is important to strengthen the relationship between universities and the national government, promote the fight against corruption, understand the security dynamics in the territory, improve planning, and implement governance models.

Concerning the university–government articulation (something that has been explored in Triple Helix models), the informants discussed the progress Medellín has made in this regard, for instance, the Ruta N innovation complex and the cooperation between higher education institutions and local government initiatives. A participant referred to this point,

*"( . . . ) I think we have made some progress with strategies to promote innovation in the municipality or such as the Ruta N industrial complex, the articulation of the municipal government and universities."* (E-01)

In Colombia, fighting corruption is a key factor because, according to different public transparency metrics, it produces a significant loss of resources that could be invested in different elements that compose a smart city model. Nevertheless, according to the participants, corruption is a structural problem that involves not only the public sector but also private agents,

*"Ending the corruption existing at different levels of society, which hinders the latter from accomplishing its objectives."* (E-02)

The informants also mentioned political challenges related to security inside the territory. This variable is part of the structure of the sociopolitical construction of the country, and, in the specific case of Medellín, it still poses challenges for its migration to smart city models. Such migration should be accompanied by adequate city planning in order to integrate local development plans, which are planning and management tools that elected officials use to define their workplan during their time in office. Likewise, governance models should be adopted to promote citizen empowerment and exercise political control.

In relation to technological challenges, the conception of smart cities as economic agents on the interconnected world stage [30] considers information technologies as drivers of innovation in complex ecosystems [27]. Hence, said technologies help to address the common problems of large cities (such as unemployment, homelessness, inequality, urban traffic, diseases, and violence) to improve the living conditions of citizens [31]. However, regarding the local context, the participants indicated that, in order to migrate to smart city models, it is difficult to ensure the accessibility and usability of technologies, consolidate innovation ecosystems, fight technology illiteracy, invest in data analysis technologies, and differentiate digital preferences and needs by age group.

Aware of the importance of the technological dimension in the migration to smart city models, the informants talked about the existing challenges in Medellín, where not only the access to new technologies but also their usability should be guaranteed. This point can be more complex if we consider the large gaps that exist between rural and urban areas in this city, which, as mentioned above, are a reflection of a structural failure of the country. In this respect, the participants commented,

*"( . . . ) if we're talking about a smart city, it's obviously a city that is significantly invested in the technological component. Obviously, it should guarantee not only the connectivity to networks but also their usability and that people appropriate them."* (E-01)

*"The concept of smart city, to me, is an idealistic one because it includes a series of adaptations to use new technologies efficiently, but it is based on the idea that all the population has access to new information technologies."* (E-03)

In line with this, another challenge discussed by the interviewees was fighting technology illiteracy. In this day and age, in which ICTs have permeated the social, economic, political, and cultural life of human beings, they must acquire technology skills. However, this represents challenges for institutionality because the development of these kinds of skills should be accompanied by accessibility to technologies; otherwise, the initiative will still represent a problem in relation to the migration to smart city models because they are supported on the use of technologies (as reported in Section 2, Theoretical Background).

The participants also indicated that to make this migration possible, innovation ecosystems should be consolidated. In their opinion, Medellín has made substantial progress in this field with investment in infrastructure that attracts new ventures that have a high growing potential, such as the case of Ruta N. Moreover, the participants suggested that the preferences and needs of the communities should be differentiated by age and gender. Data mining plays a crucial role for this purpose, and, in the current global crisis caused by COVID-19, it became more apparent, as one of the interviewees replied,

*"Sometimes you see people, even young individuals; for example, an average individual, a 40-year-old person may not like to use information and communication technologies to make transactions and report information. Not everyone does it. So, I think this question refers to one of the biggest challenges."* (E-07)

Finally, the study case faces two other difficulties to migrate to smart city models: environmental and conceptual challenges. In relation to environmental issues, the key informants commented on the relevance of fostering the sustainability of resources, promoting sustainability education, and raising environmental awareness. Environmental sustainability is considered to be an important aspect for the development of smart cities, but the discussion of this issue has a more political character, considering international resolutions and innovative solutions to combat complex urban challenges [22]. Regarding conceptual challenges, the term smart city should be adapted to the needs of each context.

The experts also considered that the intention to migrate to smart city models is largely determined by the role of public actors. Therefore, according to them, such actors have the great responsibility of assigning resources and creating policies that favor this conception of city, for example,

*"The political part plays a leading role in the implementation and resource allocation of any project. For a smart city, it is vitally important that its political class understand the importance of sustainability and smart cities, as well as what it takes to become a smart city."* (E-04)

*"( . . . ) a smart sustainable city involves many aspects. A city can be smart but not sustainable if it focuses too much on exploiting current resources and ignores the fact that the world doesn't end today, i.e., that it is necessary to offer possibilities and decent conditions to those who will come in 10, 15, 20, or 30 years. Therefore, a smart city should also be sustainable."* (E-03)

Two other factors were revealed as environmental challenges: sustainability education and raising environmental awareness. According to the participants, these factors involve not only the expression of political will by government actors (representatives of the public sector) but also active citizens that support the rational use of the available natural resources and understand that current and future generations need them.

Additionally, in relation to the conceptual challenges of a transition to smart city models, the informants highlighted the importance of adapting the concept of smart city to the needs of the context in which it will be implemented, in this case, Medellín (Colombia).

This idea is vitally important because a smart city is a complex system where multiple actors come together. Therefore, its functionality would depend on the feedback the system receives and its adaptation, and the concept of smart city should be defined in accordance with the needs of the territory in order for it to be efficiently implemented.

About this aspect, some informants said,

> *"Well, let's say there are multiple challenges. The first one is defining what is understood by 'city' and 'smart city', identifying the characteristics of each city, its populational groups, and the communities that live there because a radical, pure smart city model can't be applied in some cases. So, that's the first challenge, trying to face and differentiate those cultural characteristics so that any element of smart city to be applied is indeed reasonable for the selected community."* (E-06)

### 4.2. Key Components in a Methodology to Evaluate Smart City Sustainability

The participants in this study were asked to propose a methodology to evaluate the sustainability of smart cities, something that has become especially important because migrating to that concept of city is closely linked to the sustainable development of territories.

The interviews revealed that a methodology to evaluate the sustainability of smart cities should include five key elements: (1) defining concepts, (2) establishing guiding principles, (3) selecting the analysis approach, (4) establishing indicators, and (5) making a diagnosis (Figure 3).

The first key element in this methodology would involve clearly defining the concepts, which implies having a conceptual knowledge of smart cities and ensuring that the methodology is consistent with the global and local dynamics, as one participant commented,

> *"Well, the most important thing is the entire conceptual foundation. And I think that, to design a methodology such as the one proposed in this study, it has to be a methodology where the concepts are linked not only to global ideas. The 2030 Agenda is not the only reference point, but we should also define how to connect this to the local context."* (E-01)

The second key element, i.e., establishing guiding principles, involves ensuring comprehensiveness, the promotion of social inclusion, and the definition of a differentiated approach. This is articulated with the Sustainable Development Goals (SDGs) with regard to their vision of sustainability as transformation in environmental, economic, and social terms. In this respect,

> *"If we're talking about sustainability, we must necessarily talk about comprehensiveness because sustainability has many components: political, social, economic, and environmental aspects. Therefore, the first and most important factor should be the comprehensiveness of the methodology."* (E-04)

The third element the interviewees mentioned in the methodology was the analysis approach. They highlighted the importance of differentiating the type of system under which the smart city would operate (simple or complex) and employing mixed methods in the analysis (including quantitative and qualitative research).

This is especially relevant because although smart city models are an alternative to the smart intervention of new urbanization dynamics, they also face problems in human settlements caused by pollution, the rising demand for utilities, and social inequality, among other issues. Therefore, evaluating sustainability requires a comprehensive approach that takes into account the type of system and a mixed-method evaluation. In this regard,

*"( . . . ) the approach, for example, that you choose to design the methodology, whether quantitative or qualitative, should have indicators that can really provide an answer based on that ambivalence of an evaluation that is not only objective. Let's say that statistics has very important properties as well, but let's not forget people's subjective appraisals."* (E-01)

The fourth key element is establishing indicators. The informants claimed that it is necessary to define the indicators that describe issues from objective and subjective perspectives. In addition, such indicators should reflect the quality of life of the inhabitants of a place that aims to become a smart city.

This is important not only for the methodology discussed here but also because it can make a contribution to the 2030 Agenda for Sustainable Development (a concept that is related to smart cities). Establishing indicators is crucial to evaluate objectives and goals regarding the development of human potential. In relation to the definition of objective metrics in the methodology, one of the informants said,

*"The degree of satisfaction of said needs should be measured employing some clearly defined metrics; certainly, there should be many reference models for that purpose. Now, any methodology implemented to evaluate the sustainability of a smart city should incorporate parameters and mechanisms that can be used to objectively evaluate the aspects mentioned earlier."* (E-03)

Finally, the participants proposed a fifth element in a methodology to evaluate the sustainability of smart cities: making a diagnosis. For this purpose, the needs and particularities of each territory should be identified, the involved actors should be mapped, and the legal system of the location should be considered.

Making a diagnosis of a territory is a decisive factor in relation to the attainment of the SDGs because this tool can be used to thoroughly monitor the dynamics and transformations that communities undergo in economic, political, and social terms. In Colombia, such a diagnosis is usually based on regional development plans.

Addressing the identification of needs in the territories, some key informants commented,

*"( . . . ) Another thing that's important when we talk about methodologies is that let's say, we are reusing them from other places in the world, which is very common in our scientific environment because in many of our studies what we do is to look at what others are doing in Europe and the United States in order to apply it to our territories. Then, an important factor is that the methodology can be adequately applied to our territory."* (E-04)

*"The first factor in such methodology should be a good diagnosis to understand the needs of the city. Sometimes Medellín is compared to big cities. This is a small city, and, although it is the smartest and most innovative in Latin America, it has different needs."* (E-05)

One of the interviewees mentioned the relevance of mapping the actors involved in the dynamics of a smart city,

*"Regardless of the methodology, whether based on hard or soft systems thinking, you should consider the actors that can somehow have an influence on the phenomenon under analysis. With respect to smart city sustainability, who are the interested agents that may influence the sustainability process?"* (E-06)

Another informant referred to the political system of the territory,

*"( . . . ) the methodological steps should resemble those you follow to design a custom-made dress; you should know the size and the measurements because it must fit perfectly. In this case, it should be adapted to a context such as ours, which is constantly changing. Regulations can constantly change; for example, the law we have today may be overturned in a month, and we have to draft a new one. I think that would be a factor."* (E-01)

This second category of analysis reveals the complexity of designing a methodology to evaluate the sustainability of smart cities. This methodology should be connected to the administrative process implemented in a specific urban policy scenario, which reaffirms the importance of planning strategies.

## 5. Discussion

The sustainability of smart cities involves more than the environmental dimension; it encompasses a series of criteria that should be used to define actions and strategies for urban development, e.g., electricity and water consumption, transportation systems, waste management, security, technology inclusion, and sociocultural development, among others [32,33]. Different models can be used to evaluate sustainability; some of them focus on certain aspects more than others, which is valid depending on the field of expertise or approach under evaluation.

The concept of a smart city has become more relevant due to the current global interest in leading societies in a sustainable manner and because people's actions are having an increasing effect on the planet. The utilization of resources jeopardizes not only the sustainability of future generations but also the sustainability of the present generation, causing serious problems that go beyond the environmental sphere, and also involve social, economic, and political aspects.

For instance, different are the report approaches and dimensions that should be evaluated when sustainability and smart cities are discussed. The environmental approach integrates controlled urban sprawl strategies to affect biodiversity as little as possible, pollution controls, the rational use of resources, and renewable energy technologies. In turn, economic approaches are closely related to city development, where industrial development is fundamental for a good perception of development [34].

Institutional development and social matters are more integrated, making citizens more important and including other points of view and social and cultural dynamics that are essential to discuss sustainable and diverse development. The technological aspect has played a vital role in recent years because the new information and communication technologies are indispensable for smart cities and are a common denominator of the solution to their problems. These approaches have a clear relationship with this study, where these aspects are necessarily framed for a comprehensive evaluation of the sustainability of smart cities.

Sustainability is evaluated using quantitative and qualitative indicators on different spatial scales, i.e., from a building to a neighborhood, city, or urban area. The analysis can also include materials, energy, and air quality as individual aspects, and even transportation planning, community development, social wellbeing, governance, and innovation. Therefore, a wide range of aspects should come together to evaluate sustainability as a whole [34]. Several countries have developed different types of evaluation to produce a more balanced assessment of the environmental, social, and economic dimensions and to draw a bigger picture to analyze.

The BREEAM (Building Research Establishment Environmental Assessment Methodology) certification systems are the first of their kind regarding building construction and planning community development because they place more emphasis on environmental and sustainability issues [35]. In turn, the CASBEE-UD method, developed by the Japanese government and academy, is focused on sustainable and efficient development based on design and planning [36]. The LUD includes urban development aspects and integrates security, natural resource management, and efficiency in transport and construction [37]. The Green Building Index (GBI) emphasizes water and energy consumption, the protection of natural ecosystems, the development of transportation systems, employment generation, and entrepreneurship. The IGBC, developed in India includes, in addition to city planning and land use, innovation and technology [38]. Note that many of these tools, used around the world to evaluate sustainability, have different approaches. Most of them present similarities with respect to the technical aspects of space planning, construction,

and design, as well as purely environmental aspects as part of green indicators, but few include social or political aspects as fundamental components of sustainable development. Hence, it is important to have tools that integrate these aspects in order to support cities and their development (an essential part of their growth) based on economic and social factors. Such tools should bring together sustainability and more attention to the dynamics of the population and its well-being [39].

In the interviews, many of the experts agreed that Medellín should work toward the satisfaction of basic needs, which include everything related to decent housing, access to utilities, average educational attainment, households covered by basic health insurance, and employment that enables individuals to meet many of said needs. This can be difficult to achieve, but the first step is articulating all these components. In addition, we should not forget the technological component because it enables governments to manage data and companies and citizens to control their resources, all in an effective and safe manner.

Smart cities usually focus on technological development and innovation (central points that determine their classification as smart) and favor the use of technologies to deal with different problems and offer opportunities [40]. Technologies are part of the construction and design process of smart cities and are used to provide services to citizens [41]; furthermore, they may generate sustainability. Although some authors disagree with the idea that sustainable development is a consequence of the implementation of new technologies [42], others claim that the latter can address important issues, such as access to education, improved information systems, and transportation, as well as security solutions that promote human capital training and create jobs [40,43,44]. In another context it is concluded that aspects such as income level, educational attainment, and location in Latin America (specifically) are positively related to the level of city smartness. Thus, cities in countries where information and communication technologies are more developed attain higher levels of city smartness [45]. Employment can also be integrated to improve this level because it is important for the sustainable growth and progress of cities.

Currently, there is no satisfactory explanation of the transformation of cities into smart cities, but it implies commitment and prospective vision to be successful. It also requires time and the characterization of efficient governance models that include a public–private contribution as well as that of citizens. In addition, this transformation involves dimensions that cities can improve to be more efficient and competitive, facilitating new synergies and ensuring the interoperability of their operations and services, which results in a better quality of life for their inhabitants.

Therefore, we excluded eight variables: (1) inclusive innovation and ICTs, (2) competitiveness and territory, (3) energy, (4) social cohesion, (5) environment, (6) water and gas, (7) e-commerce, and (8) communications. They were not excluded because they are not important but because, in this study, we selected different variables to create a more detailed profile of the city under analysis here. As a result, the evaluation we propose is focused on six dimensions: (1) government, (2) mobility, (3) sustainability, (4) people, (5) economy, and (6) quality of life.

Although the definition of smart city has been widely disseminated and studied in recent years, a great deal of the progress in this area has been theoretical and has been made by the adoption of qualitative approaches. In more practical terms, smart cities require technology in the form of network infrastructure, which translates into a connected, data-driven society. In addition, they require technological mechanisms to improve competitiveness and support their communities, thus generating a better quality of life for their residents. Therefore, future research should examine more real and specific aspects of the dynamics of cities. This study contributes to the field because it proposes a methodological model (adapted to the dynamics of Medellín as the study case) that includes social and political aspects. This is noteworthy because most existing tools focus on technical, environmental, transportation, planning, and technological factors and do not prioritize social and political dynamics as our model does.

## 6. Conclusions

This paper makes a relevant and necessary contribution to the field because its object of study is in Colombia, which presents limitations in access to and usability of new technologies (a key aspect to adopt smart city models) but is rich in natural resources that sometimes even worsen the socioeconomic conditions of its communities when legal and illegal actors fight for them (e.g., mining).

A methodology to evaluate the sustainability of smart cities should employ not only quantitative but also qualitative tools because mixed-method approaches can offer a more comprehensive picture of the phenomenon. Likewise, to be successful, such a methodology should define indicators (to measure the accomplishment of the goals established in smart city models) and, no less important, include a diagnosis of the territory. The latter is necessary because even similar contexts may have particular needs that require a highly specific diagnosis; for instance, Bogotá and Medellín are both capital cities in Colombia, but their security and coexistence dynamics are different.

This study revealed that this methodology should not only consider the components of a smart city but also a series of challenges that justify its fifth key element, i.e., a diagnosis of the territory, which is essential to ensure the effectiveness of the proposed methodology and, therefore, its consistency.

The informants referred to social, economic, political, technological, environmental, and conceptual challenges that are decisive for the success of such methodology. This reflects the importance of other fields in the concept of the smart city, for instance, the systemic approach to processes. As a result, migrating to smart city models requires elected officials (who are key in the adoption of this concept of city) to take into account the interaction of several subsystems that determine the success or failure of following this new trend in urban development.

Smart cities require technology that involves cabling, which translates into a wired society. In addition, they need tools to improve competitiveness and support communities, thus generating a better quality of life for their inhabitants.

**Author Contributions:** In this order of ideas, A.V.-A. participated with conceptualization, methodology, validation, data curation, writing the original draft preparation and with the project administration. M.L.U.-M. supported with conceptualization, methodology, validation, data curation and writing the original draft preparation. And finally, L.B.-P. participated with the methodology, validation, data curation and writing the original draft preparation. All authors have read and agreed to the published version of the manuscript.

**Funding:** This research received no external funding. The APC was funded by Instituto Tecnológico Metropolitano (Medellin, Colombia). We would like to thank ITM Translation Agency for language editing the original manuscript.

**Institutional Review Board Statement:** The study was conducted according to the guidelines of the Declaration of Helsinki, and approved by the Ethics Committee of Corporación Universiraria Americana (protocol code 07/21.03.2020).

**Informed Consent Statement:** Informed consent was obtained from all subjects involved in the study.

**Data Availability Statement:** Not applicable.

**Conflicts of Interest:** The authors declare no conflict of interest.

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
