# Peer review of "A Methodological Model to Evaluate Smart City Sustainability"

_sustainability, doi:10.3390/su132011214_

Round 1

Reviewer 1 Report

It is an interesting study with innovative ideas, although the basis is not very strong (a few questionnaires from -key?- sources). Perhaps the paper can be improved slightly by reconsidering (and targeting better) the relationship of smartness and sustainable development, for example in Fig. 2 .

If you can also consider relevant key international literature it can be improved. It has to relate to broader issues, not only Medellin specificities, so it can be interesting to broader audience.

Reviewer 2 Report

Dear authors,

An interesting paper on smart cities, based on  Medellín, Colombia.  I have the following points with relevant line numbers for you to consider:

95-105 include issue of waste (plastic etc) and also extend 
conclusion on SD beyond anthropogenic focus to integrity of biosphere?
153 more detail on search methodology "we reviewed the scientific literature" 
e.g. approach, search terms, engines, databases?
156 criteria for selection?
164 details on 'key informants' for SSI instrument review?
186 error SPACE "nonmaleficence"?
542+ acronyms e.g. BREEAM spelt out?
621 last key variable 'context'  but this did not feature earlier when you listed variables?
629 notion of "cabling" suddenly introduced.  The whole conclusion 620-631 seems fragmented and needs tightening for synthesis, perhaps involving “the bottom-up or citizen-led approach”?

Extra references, e.g. 

AJ Scott (2008) Resurgent metropolis?
AC Pratt (2008) Creative cities
https://www.theguardian.com/cities/2014/dec/17/truth-smart-city-destroy-democracy-urban-thinkers-buzzphrase

I hope my suggestions are useful and help improvement of your paper.
